# Dynamics of the perception and EEG signals triggered by tonic warm and cool stimulation

Dounia Mulders[1,2]*, Cyril de Bodt[1], Nicolas Lejeune[2], Arthur Courtin[2], Giulia Liberati[2], Michel Verleysen[1], André Mouraux[2]

**1** ICTEAM institute, Université Catholique de Louvain, Louvain-la-Neuve, Belgium, **2** IONS institute, Université Catholique de Louvain, Brussels, Belgium

* dounia.mulders@uclouvain.be

**Data Availability Statement:** The data used for the manuscript are publicly available from the OSF data repository at the address: https://osf.io/q45u8/ (DOI: 10.17605/OSF.IO/Q45U8).

## Abstract

Thermosensation is crucial for humans to probe the environment and detect threats arising from noxious heat or cold. Over the last years, EEG frequency-tagging using long-lasting periodic radiant heat stimulation has been proposed as a means to study the cortical processes underlying tonic heat perception. This approach is based on the notion that periodic modulation of a sustained stimulus can elicit synchronized periodic activity in the neuronal populations responding to the stimulus, known as a steady-state response (SSR). In this paper, we extend this approach using a contact thermode to generate both heat- and cold-evoked SSRs. Furthermore, we characterize the temporal dynamics of the elicited responses, relate these dynamics to perception, and assess the effects of displacing the stimulated skin surface to gain insight on the heat- and cold-sensitive afferents conveying these responses. Two experiments were conducted in healthy volunteers. In both experiments, noxious heat and innocuous cool stimuli were applied during 75 seconds to the forearm using a Peltier-based contact thermode, with intensities varying sinusoidally at 0.2 Hz. Displacement of the thermal stimulation on the skin surface was achieved by independently controlling the Peltier elements of the thermal probe. Continuous intensity ratings to sustained heat and cold stimulation were obtained in the first experiment with 14 subjects, and the EEG was recorded in the second experiment on 15 subjects. Both contact heat and cool stimulation elicited periodic EEG responses and percepts. Compared to heat stimulation, the responses to cool stimulation had a lower magnitude and shorter latency. All responses tended to habituate along time, and this response attenuation was most pronounced for cool compared to warm stimulation, and for stimulation delivered using a fixed surface compared to a variable surface.

## Introduction

Cutaneous thermosensation contributes to maintaining core body temperature, to the discrimination of manipulated object properties based on its thermal characteristics and, importantly, to the detection of potentially harmful cold and heat stimuli [1, 2]. The qualitatively-different sensations elicited by thermal stimuli result from the activation of a variety of cold-

**Funding:** DM, CdB and NL are Research Fellows of the Fonds de la Recherche Scientifique - FNRS (https://www.frs-fnrs.be/fr/). AC is supported by a FRIA doctoral grant from the FNRS. GL is a FNRS researcher. The funders had no role in study design, data collection and analysis, decision to publish, or preparation of the manuscript.

**Competing interests:** The authors have declared that no competing interests exist.

and heat-sensitive thinly myelinated Aδ and unmyelinated C fiber thermonociceptors, having conduction velocities in the range of 3 – 30 m/s and around 1 m/s, respectively [3]. Thermonociceptors can be further classified according to the types of stimuli to which they respond, and the way they respond to these stimuli [4]. A given thermonociceptor can be activated by a particular range of temperatures and, in the case of a polymodal nociceptor, can also be activated by mechanical or chemical stimuli [1, 5]. When thermal stimuli are maintained over time, the response of thermonociceptors can adapt either quickly or slowly [1, 4, 6]. Furthermore, repeated activation of thermonociceptors can induce activity-dependent slowing, which is most pronounced in C fibers, but also varies across different types of C-fiber afferents [7, 8]. The molecular basis for thermal transduction by thermonociceptors involves a diversity of cold- and heat-sensitive Transient Receptor Potential (TRP) ion channels [2]. Despite remarkable advances in the understanding of how thermonociceptors respond to thermal stimuli [2, 5], the links between activation of some populations of thermonociceptors, elicited brain activity and subsequent human thermal perception are not yet fully elucidated. In order to delineate these links, there is a need to characterize and compare perception and brain responses to a large variety of thermal stimuli, by varying their features such as their intensity and duration.

Several studies already aimed at preferentially activating specific types of thermosensitive afferents, mainly using electroencephalography (EEG) and the recording of event-related brain potentials (ERPs) [9]. For example, brief laser heat stimuli applied onto the skin generate laser-evoked brain potentials (LEPs) that are thought to mainly result from the activation of one class of polymodal nociceptors: so-called 'Type 2' (quickly-adapting) mechano- and heat-sensitive Aδ fiber nociceptors (AMH-2) [10, 11]. Although such stimuli are also expected to activate quickly-adapting C fiber thermonociceptors, co-activation of these unmyelinated afferents does not elicit any evident brain activity compatible with their slow conduction velocities [12, 13]. Conversely, similar short-lasting laser heat stimuli with a lower intensity can elicit LEPs whose latencies are compatible with the conduction of unmyelinated C fibers, probably because such stimuli selectively activate heat-sensitive C fiber thermonociceptors [14]. More recently, it was shown that rapid innocuous cooling of the skin elicits cool-evoked brain potentials whose short latencies are compatible with the activation of cool-sensitive Aδ fiber afferents [15–17].

Because the responses elicited by very transient thermal stimuli can be expected to predominantly reflect activity generated by quickly-adapting thermonociceptors responding vigorously to rapid changes in skin temperature, not much is known about the brain responses elicited by tonic thermonociceptors. Mainly using functional magnetic resonance imaging (fMRI) [18–20] or positron-emission tomography (PET) [21, 22], some previous studies characterized the brain activity elicited by long-lasting heat stimuli inducing tonic heat pain over several seconds. However, the low temporal resolution of fMRI and PET makes it difficult to tease out brain activity elicited by quickly-adapting thermonociceptors responding at the onset of the thermal stimulus from activity elicited by slowly-adapting thermonociceptors responding gradually and in a sustained fashion over time. Other studies have attempted to identify sustained changes in ongoing oscillatory activity related to the perception of tonic pain using EEG [23–27]. One interesting approach that was attempted using both EEG and fMRI is to introduce random variations in the intensity of the eliciting stimulus (or exploit the spontaneous pain fluctuations) in order to relate the time course or temporal structure of the measured brain activity (e.g. fluctuations in the fMRI-BOLD signal, or fluctuations in the magnitude of ongoing EEG oscillations) with the time course or temporal structure of tonic pain perception over time [19, 24, 28]. Lately, Colon et al. proposed a new method to characterize the cortical processes related to tonic heat perception, based on 'EEG frequency tagging' using long-lasting sinusoidal heat stimulation of the skin [29]. Periodic modulation of a sensory stimulus can be

expected to elicit synchronized periodic activity in the neuronal populations responding to the stimulus, sometimes referred to as a steady-state response (SSR) [30]. Unlike ERPs, SSRs are thus sustained over time. In the frequency domain, the stimulus-evoked 'frequency-tagged' activity concentrates at the frequency of stimulation and its harmonics, and is thus easy to isolate from non-stimulus-related activity. In the time domain, the known periodicity also allows highlighting the dynamics of the elicited responses. The recording of SSRs with EEG is proposed to complement the classical approach of recording ERPs [31–33]. Most importantly, depending on the frequency of stimulation and, possibly, the shape of the periodic stimulus, periodic thermal stimulation can be expected to preferentially activate different types of thermonociceptors. For example, periodic radiant heat stimulation of the skin to 50˚C at a very slow oscillation frequency of 0.2 Hz was shown to generate a periodic EEG response predominantly conveyed by unmyelinated C fibers [29].

In the present study, using a novel contact thermal stimulator that allows generating well-controlled cooling and warming ramps [16], we characterize for the first time SSRs related to the sustained periodic activation of cool-sensitive afferents, compare these responses to the SSRs elicited by periodic noxious heat stimulation, and relate the temporal dynamics of the elicited SSRs to the temporal dynamics of the stimulus-induced cold and heat sensations. As explained above, sustained or repeated stimulation of thermonociceptors can be expected to induce some amount of peripheral habituation [34] and/or activity-dependent slowing. In addition, central mechanisms can also affect the perception and brain responses to sustained stimuli [35]. In particular, sensitization mechanisms, wind-up or temporal summation [36, 37] could lead to enhanced responses along time, while habituation mechanisms or more abrupt phenomena such as offset-analgesia [38] could decrease the observed responses. Therefore, taking advantage of the fact that the thermal stimulator allows separately controlling five different zones of the probe contacting the skin, we also assessed the effect of displacing the stimulated skin area across the stimulation cycles. All stimuli consisted in a sinusoidal temperature profile oscillating at 0.2 Hz between a neutral temperature of 31˚C and either 14˚C (cool stimulation) or 48˚C (heat stimulation). The amplitude of the cool and heat stimuli was thus identical ($\Delta = 17$˚C), but the cool stimulation was expected to generate activity within cool-sensitive afferents whereas the heat stimulation was expected to generate activity within heat-sensitive nociceptors. In a first experiment, we collected continuous subjective intensity ratings from healthy subjects exposed to the warm and cool stimuli applied to a fixed or varying area of the skin along the stimulation cycles. In a second experiment, we recorded the EEG of healthy subjects exposed to the same stimuli, allowing to confront the observed EEG features with subjective perception. The baseband EEG responses and ongoing oscillations in physiological frequency bands were analyzed in the time, frequency and time-frequency domains.

## Materials and methods

### Participants

Two groups of healthy volunteers took part in Experiment 1 (5 men and 9 women, aged 24-35 years) and Experiment 2 (6 men and 9 women, aged 21-34 years), respectively. All participants were right-handed and did not suffer from any neurological disorder. The study was approved by the local ethics committee (Comité d'Ethique Hospitalo-Facultaire de l'Université catholique de Louvain, B403201316436) and all participants gave written informed consent.

### Contact thermal stimulation

All the stimuli considered in this study were delivered using a prototype contact thermal stimulator (TCS-II) made of micro Peltier elements whose temperature can be varied at rates of up

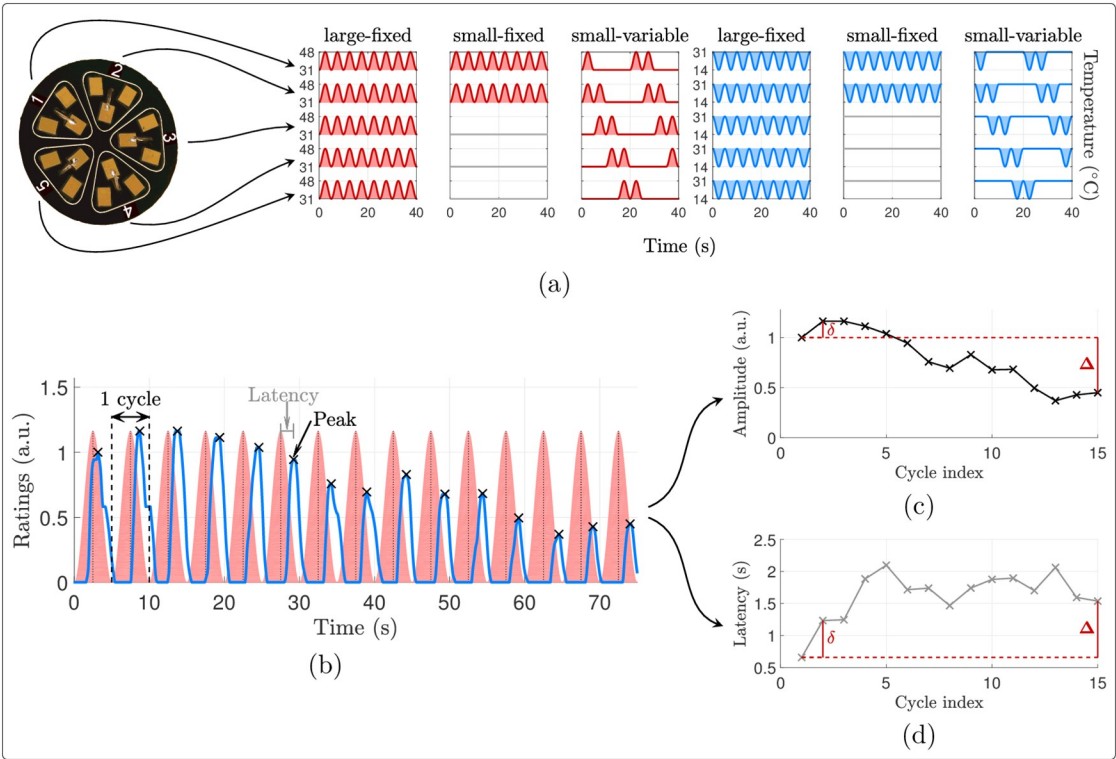

**Fig 1. Stimulation employed and intensity ratings collected.** (a) Stimulation surface of the Thermal Cutaneous Stimulator (TCS) employed and stimulation temperature profiles considered in Experiment 1, depicted in red (or blue) for the warm (or cool) stimuli and truncated to 40 seconds for readability. The temperature profiles of the small-fixed and small-variable conditions were also employed in Experiment 2. (b) Example of intensity ratings from one subject during one stimulus (warm large-fixed). The temperature waveform is shaded in red and the blue curve is the intensity rating. In each cycle of 5 seconds, the amplitude of the rating peak and its latency compared to the corresponding temperature peak were computed. (c-d) Definition of the first and final differences, denoted respectively by $\delta$ and $\Delta$, of (c) the rating peaks amplitudes and (d) latencies along the cycles.

to 300°C/s (QST Lab, Strasbourg, France). The stimulation probe weights 440 g and has a flat 30-mm diameter surface which is applied against the skin and contains 15 micro Peltier elements, as indicated in Fig 1a. The Peltier elements are organized in five zones of around 24 mm². The temperature is controlled independently in each zone, allowing to vary the stimulated skin surface without displacing the stimulation probe. The temperature of the probe is negligibly affected by the subject's skin temperature [16].

All stimuli consisted in 15 periods of 0.2 Hz sinusoidal cooling (between 31 and 14°C) or warming (between 31 and 48°C) of the skin. Each stimulus thus lasted 75 seconds (15 cycles of 5 seconds). The maximal warm temperature of 48°C was chosen such as to recruit the largest amount of heat-sensitive thermonociceptors while avoiding burn lesions due to stimulus duration [6]. The frequency of 0.2 Hz was selected because previous studies showed that, at this frequency, radiant heat stimuli generate clear SSRs related to the activation of heat-sensitive C-fiber afferents [29]. The same frequency, stimulus duration and amplitude of temperature variation was used for cool stimulation, such as to allow a direct comparison of heat and cool-evoked SSRs.

## Experiment 1: Time course of heat and cool perception

The goal of this first experiment was to assess the time course of the perception elicited by long-lasting periodic cool and warm stimuli oscillating at a frequency of 0.2 Hz. In order to

study the effect of displacing the stimulation on the skin surface, three types of stimulation profiles were used for both cool and warm stimulation: (1) synchronous activation of all five zones of the probe (large-fixed), (2) two alternating zones per cycle (small-variable), and (3) two fixed zones during the entire stimulation (small-fixed). We chose to stimulate using 2/5 zones of the probe because the sensation elicited by activating only one zone was very weak. The temperature profiles are illustrated in Fig 1a. Two epochs were delivered for each of the six conditions, in a randomized order across participants and with self-paced inter-stimulus intervals, on either the right or left volar forearm. The stimulated arm was resting on a soft surface with the volar side upwards. The probe was manually displaced after each 75-s stimulus to avoid trial-to-trial habituation or sensitization. Due to technical problems, one participant could not receive the stimuli from the cool large-fixed condition.

During stimulation, the subject was asked to continuously rate the perceived thermal intensity on a visual analog scale (VAS). The participant had to displace a 10 cm vertical slider with the contralateral hand. The extremities of the slider were defined as lowest and highest intensities that can be imagined. The continuous ratings were digitized at 1000 Hz with an analog/digital converter (USB-6343, National Instruments, Texas) and the two epochs of each condition were then averaged. Furthermore, at the end of each 75-s stimulus, subjects were asked to describe the quality of the sensation by selecting one or more descriptors from the following list: 'not perceived', 'light touch', 'touch', 'tingling', 'pricking', 'warm', 'hot', 'burning', 'cool', 'cold' and 'humid' [10]. These reports lead to an average inter-stimulus interval of 50 seconds and the experiment lasted approximately 20 minutes.

To study the time courses of the intensity ratings and how they were affected by the temperature (warm or cool) and the way the stimulation probe was employed, some features were extracted and analyzed as follows. For all statistical tests mentioned, significance level was set to 5%. When multiple comparisons were performed, this level was adapted with the Holm-Bonferroni correction, as detailed in what follows.

**Average features.**   For each intensity rating waveform, per subject and type of stimulus, the point of maximum rating was identified in each of the 15 stimulation cycles. These maximum ratings and their latencies relative to the stimulation cycle (0 second corresponding to when the temperature change relative to baseline was maximal) were averaged across stimulation periods. These two measures were then compared across stimulation conditions using a two-way repeated measures ANOVA with the factors temperature (cool or warm) and surface (large-fixed, small-fixed, small-variable). The numbers of degrees of freedom of the $F$-distributions were adapted with the Huynh-Feldt correction when the condition of sphericity was violated (according to Mauchly's test for sphericity) [39, 40]. Whenever the effect of one of the factors was significant, post-hoc paired sample $t$-tests were conducted with Holm-Bonferroni correction [41]. Furthermore, for each type of stimulus, the relative delay between the maximum intensity rating and stimulation was assessed by comparing the average latencies of maximum rating against zero using one-sample $t$-tests.

**Temporal dynamics of heat and cool perception across stimulation cycles.**   Each rating time course was then normalized such that the maximum and minimum ratings during the first stimulation cycle corresponded to 0 and 1 respectively, as illustrated in Fig 1b. This allowed characterizing the temporal dynamics of intensity ratings along the stimulation cycles without being affected by initial differences in rating amplitude across conditions. These waveforms normalized to the ratings of the first stimulation cycle were used to extract the maximum rating of each stimulation cycle (expressed relative to the rating of the first stimulation cycle) and its latency. Within these normalized ratings, if the intensity of the stimulus-evoked sensation tended to increase (resp. decrease) across stimulation cycles, this would result in maximum ratings becoming greater (resp. smaller) than 1. Since the evolution of the intensity

of a long-lasting sensation is typically nonlinear, with an early strong decrease (or increase) rate when habituation (or sensitization, respectively) occurs [35, 36, 42, 43], the changes in maximum ratings and in latencies of the maximum ratings across stimulation cycles were characterized both between the first and the second cycle (to assess immediate changes in perception already occurring after the first cycle), and between the first and the last cycle (to assess global changes in perception occurring across the 15 cycles). The immediate and global changes in rating intensities and latencies, denoted respectively by $\delta$ and $\Delta$, are illustrated in Fig 1c and 1d. These quantities were compared across stimulation conditions using two-way repeated measures ANOVAs with the factors 'temperature' (cool or warm) and 'surface' (large-fixed, small-fixed or small-variable) as within-subject fixed factors. Post-hoc paired $t$-tests were conducted when justified. In addition, one sample $t$-tests were employed to assess the significance of each of these changes against 0.

Other features of the percept elicited by periodic cool and warm stimulation were assessed, such as the minimum ratings of intensity across stimulation cycles and the amplitude of the cyclic variation in ratings of intensity. These are reported as Supplementary Material.

## Experiment 2: Electroencephalographic recordings

In this second experiment, the EEG was recorded while participants were exposed to small-surface cool and heat stimuli delivered using two zones of the contact probe at either a fixed or variable skin location, resulting in four conditions: cool-variable, cool-fixed, warm-variable and warm-fixed. Participants were seated with their right forearm resting on a soft surface, with the volar side upwards. They were instructed to keep their gaze fixed at eye level. All stimuli were delivered to the right volar forearm. Such as in Experiment 1, each stimulus lasted 75 seconds (15 periods of a 0.2 Hz sinusoidal waveform). Each type of stimulus was repeated 12 times, presented in a randomized order. Inter-stimulus interval was self-paced by the experimenter and varied between 10 and 20 seconds. The TCS-II probe was manually displaced after each stimulus. The entire recording session lasted approximately 1h30. Such as in experiment 1, a significance level of 5% was used for all statistical tests, and the Holm-Bonferroni method was used to correct for multiple comparisons.

**EEG recording and preprocessing.** The EEG was recorded using 64 Ag-AgCl electrodes, whose impedances were kept below 10kΩ, placed on the scalp according to the international 10/10 system (WaveGuard 64-channel cap; Advanced Neuro Technologies). Signals were amplified and digitized at 1000 Hz, with an average reference. The EEG recordings were analyzed offline using Matlab R2017a (The MathWorks), and the preprocessing was performed using Letswave 6 (http://letswave.org) [44]. All signals were high-pass filtered above 0.05 Hz to remove slow drifts with a 4$^{th}$ order zero-phase Butterworth filter, and power line noise was removed with a 50 Hz notch filter. The epochs were defined by segmenting the EEG from 0 to 75 seconds after each stimulation onset. Each epoch was then centered, and stereotyped artifacts (eye movements or blinks and muscle artifacts) were removed using an Independent Component Analysis (ICA) decomposition [45]. For each subject, the full rank data matrix was decomposed using 63 independent components [46]. Epochs containing large artifacts were finally rejected by visual inspection, leading to exclude an average (± standard deviation) of 0.6±0.74, 0.8±1.21, 0.73±0.88 and 1.2±1.15 epochs for the conditions warm-fixed, warm-variable, cool-fixed and cool-variable, respectively.

**Frequency domain analysis.** Average waveforms were computed for each subject and condition. Since the stimulation was periodic, a periodic stimulus-evoked response was expected. This response was not likely to be a perfect linear mapping of the applied sinusoidal stimulation intensity, but could be any periodic signal with the same fundamental period of 5

seconds. The power of such a periodic signal is concentrated at its fundamental frequency of 0.2 Hz and its harmonics (0.4, 0.6, 0.8, . . . Hz) [47, 48]. To identify the presence of a stimulus-evoked EEG response, the Fourier Transform (FT) of each 75-s average waveform was hence computed per subject, stimulation condition and electrode with a frequency resolution of 0.013 Hz. The significance of the signal amplitude at the stimulation frequency and its first harmonics was then assessed. To do so, the noise level at each frequency of interest (FOI) was estimated using the average amplitude at eight neighboring frequencies (located at ±{0.027, 0.04, 0.053, 0.066} Hz around each FOI) and was removed from the spectrum, resulting in *noise-subtracted (NS) amplitudes* [31, 48]. The significance of the NS amplitudes at the FOI $f_k$ = $k \cdot 0.2$ Hz, for $k$ = 1, . . ., 5, was then tested using one sample $t$-tests against 0. The electrode where the NS component at 0.2 Hz was the largest on average across conditions was then selected for the next analyses. Then, the NS amplitudes at the stimulation frequency were compared across stimulation conditions using a two-way repeated measures ANOVA with the factors 'temperature' (cool or warm) and 'surface' (fixed or variable). Post-hoc paired $t$-tests were conducted when justified.

**Time domain analysis.** The 75-s average waveforms were further segmented in 15 periods of 5 seconds and averaged to analyze the stimulus-induced periodic EEG waveform and assess its latency across the different conditions. Response latency was defined as the difference between the peak of the EEG response and the latency of maximum temperature change. Latencies were compared across conditions using a two-way repeated measures ANOVA with the factors 'temperature' (cool or warm) and 'surface' (fixed or variable). Paired $t$-tests were used for post-hoc comparisons. Furthermore, for each type of stimulus, the relative delay between the EEG response and stimulation was assessed using one-sample $t$-tests against 0.

**Time-frequency analysis.** As each stimulus lasted 75 seconds, the stimulus-evoked EEG signals cannot be assumed to be strictly stationary along such a long time interval. Time-frequency analysis of the standardized recordings averaged across the epochs were therefore considered in order to highlight the temporal dynamics of the EEG components at 0.2 Hz. Short-time Fourier transforms (STFTs) were employed to characterize the frequency content of the recordings at frequencies ranging from 0.02 Hz to 1 Hz in steps of 0.02 Hz. The STFT was preferred to a continuous wavelet transform as the time resolution of interest was determined by the stimulation period, and was therefore fixed across frequencies. By analogy with the complex Morlet wavelet which is appropriate to characterize time-frequency components of EEG signals [49–51], the STFT was implemented using a full-length Gaussian window. The employed STFT of a signal $x$ is hence defined as

$$\Psi_x(\tau, f) = \int_{t=-\infty}^{+\infty} x(t) \cdot \exp\left(\frac{-(t-\tau)^2}{2 \cdot (L_w/6)^2}\right) \cdot \exp\left(-2\pi j f(t-\tau)\right)dt. \qquad (1)$$

The width parameter $L_w$ determine the number of oscillations of the modulated complex exponential, which was fixed to 5 with $L_w$ = 5, as suggested in [52, 53] to reach a good time-frequency resolution for the frequency of interest of 0.2 Hz. The STFT of the EEG recordings at the electrode with maximal SSR was computed for each subject and condition.

In order to assess the magnitude of the EEG response elicited by the periodic thermal stimuli along time, the STFT amplitudes $|\Psi_x(\tau, f)|$ (per subject and condition) were first noise-subtracted. The noise at each frequency along time was estimated, similarly to the Fourier transform, by computing the mean amplitude of the STFT at four neighboring frequencies, located at ±{0.04, 0.06} Hz around the considered one. Within these noise-subtracted time-frequency maps $\Psi_x^{NS}(\tau, f)$, the sum of the STFT amplitudes at the frequency of stimulation and

its four first harmonics,

$$\sum_{k=1}^{5} \Psi_x^{\mathrm{NS}}(\tau, k \cdot 0.2), \tag{2}$$

was used as a measure of the time course of the EEG response elicited by the periodic stimuli. The amplitude of this component at 0.2 Hz, which should not be significantly different from 0 when there is no stimulus-related activity, was compared against 0 with one sample $t$-tests along time. The global response amplitude was then defined as the area under the curve (AUC) of this component. This AUC was compared across temperatures (cool or warm) and surfaces (fixed or variable) with a repeated measures ANOVA and post-hoc paired $t$-tests.

Finally, to further assess the habituation of the components at 0.2 Hz across cycles, we computed the noise-subtracted FT amplitude at 0.2 Hz by progressively removing an increasing number of periods at the beginning of the signals, while keeping the signal length fixed with zero-padding. One sample $t$-tests with Holm-Bonferroni correction were used to assess the significance of the noise-subtracted amplitudes at 0.2 Hz for each number of initial cycles removed, allowing to highlight whether EEG responses at 0.2 Hz maintained after a few cycles, even if their amplitude was reduced.

**Modulation of ongoing oscillations.** Periodic sensory stimulation can also be expected to induce a periodic modulation of the magnitude of ongoing EEG oscillations [29, 54]. In order to assess whether periodic 0.2 Hz cool and warm stimulation delivered using a fixed or variable surface induced a periodic modulation of ongoing EEG oscillations within different frequency bands, we estimated the signal envelopes within theta (4-8 Hz), alpha (8-12 Hz), beta (12-30 Hz) and gamma (30-50 Hz) frequency bands, as follows. First, we band-pass filtered all the unaveraged EEG epochs within each of these bands using a 4[th] order zero-phase Butterworth filter and then computed the envelopes of these signals as the norms of the corresponding analytic signals that were obtained through Hilbert transforms. These envelopes were then studied in the same way as the original signals, following the procedures described above. The signal envelopes were averaged across trials, the FTs were computed and the electrode with the largest NS amplitude at 0.2 Hz was selected. Then, the signals were averaged across each stimulation cycle and STFTs were computed. The gamma range was chosen so as to avoid considering the 50 Hz frequency around the center of the band, in which case the low-frequency modulations of this band would have been hindered by the 50 Hz notch filter [55].

## Results

### Experiment 1: Time course of heat and cool perception

The time courses of the percepts elicited by the different warm and cool stimuli and averaged across the stimulation cycles are shown in Fig 2a. The grand average time courses along the cycles are depicted in Fig 3a. In the latter figure, the maximum amplitude within the first stimulation cycle does not necessarily reach 1 although the ratings were normalized, because the extrema are not perfectly time-locked across subjects. The outcomes of the ANOVAs assessing the effects of stimulation temperature and surface on intensity ratings are summarized in Table 1. Subsequent post-hoc comparisons are illustrated in Figs 2 and 3 and the results are detailed hereunder.

**Average features.** The intensity of the percept elicited by periodic warm stimulation was greater than the intensity of the percept elicited by periodic cool stimulation, regardless of the stimulation surface employed (Fig 2b and 'Mean peak' in Table 1). The latency of the sensation elicited by cool stimulation was significantly shorter than the latency of the sensation elicited

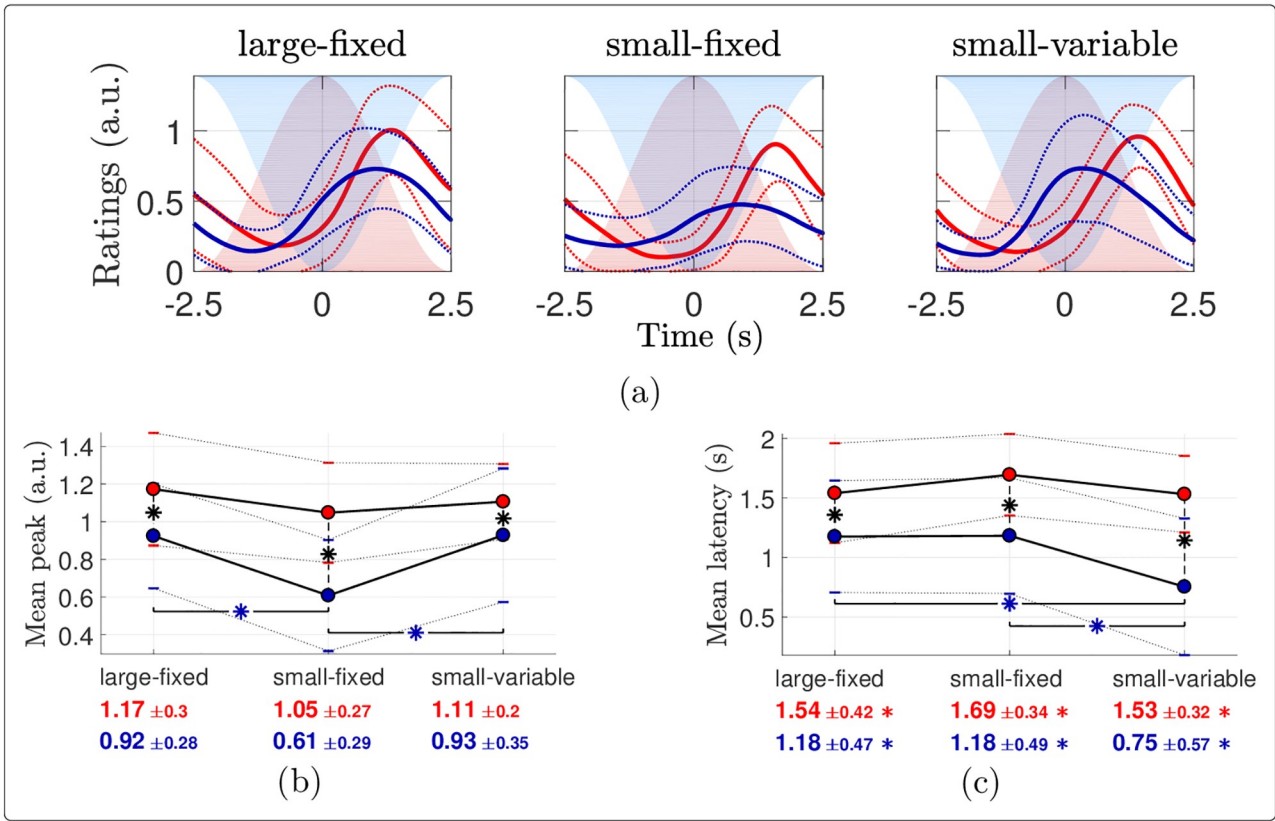

**Fig 2. Intensity ratings averaged across stimulation cycles.** (a) Group-level average (± standard deviation in dotted) time courses of the intensity ratings averaged across stimulation cycles, in (light) red and (dark) blue for the warm and cool conditions respectively. The stimulation temperatures are shaded. (b-c) Pairwise comparisons of (b) the mean peak intensity rating and (c) the mean peak intensity latency. The red (resp. blue) dots show the mean features for the warm (resp. cool) stimulation, the standard deviations being indicated with horizontal bars. These means and standard deviations are also reported below the plots with the corresponding color. Each asterisk in the plot indicates a significant difference according to paired samples *t*-tests with Holm-Bonferroni correction, in red, blue (horizontally) or black (vertically) respectively when the two compared conditions are warm, cool or different. In (c), an asterisk besides an x-axis tick label shows that the corresponding mean feature is significantly different from 0 based on one sample *t*-tests with Holm-Bonferroni correction.

by warm stimulation (Fig 2c and 'Mean latency' in Table 1). Furthermore, for cool stimulation, the average intensity of the percept was lower when stimuli were delivered using a fixed surface as compared to a variable surface (small-fixed vs. small-variable; Fig 2b and Table 1). Cool stimulation using a fixed surface also increased the latency of the maximum rating as compared to stimulation using a variable surface (small-fixed vs. small-variable; Fig 2c and Table 1).

**Temporal dynamics of heat and cool perception across stimulation cycles.** Both for cool and warm stimulation, and regardless of whether stimulation was applied using a fixed vs. a variable surface, the amplitude of the cyclic variations in ratings induced by the periodic stimulus tended to decrease along the stimulation cycles (Fig 3b, the exact reductions and their significance being indicated at the bottom of Fig 3f). This habituation of perception appeared to be stronger for cool stimulation as compared to warm stimulation, especially when stimulation was delivered using a fixed surface. The individual time courses of the rating peaks and their latencies are reported in S1 and S2 Figs.

**Immediate changes in perception occurring after the first stimulation cycle.** As shown in Fig 3d, there was no marked change in the intensity of the percept elicited by warm

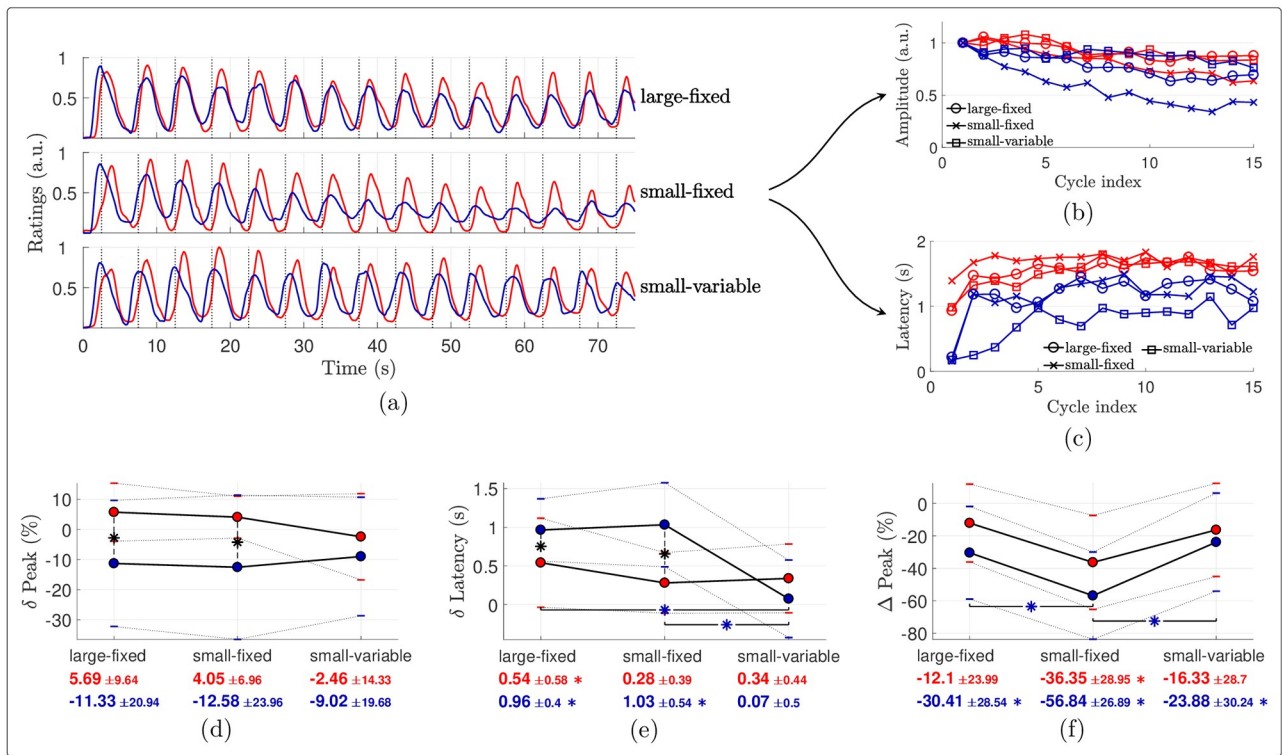

**Fig 3. Intensity ratings dynamics.** (a) Group-level average intensity ratings along the cycles, in (light) red and (dark) blue for the warm and cool conditions respectively. The vertical dotted lines indicate the times of maximum temperature change (2.5 s after the beginning of each stimulation cycle). From each normalized individual time course, the rating peaks and their latencies within each cycle are extracted, their grand average being illustrated in (b) and (c). (d-e-f) Pairwise comparisons of (d) the early change in peak intensity rating, (e) the early change in peak latency and (f) the global change in peak intensity rating. The red (resp. blue) dots show the mean features for the warm (resp. cool) stimulation, the standard deviations being indicated with horizontal bars. These means and standard deviations are also reported below the plots with the corresponding color. Each asterisk in the plot indicates a significant difference according to paired samples *t*-tests with Holm-Bonferroni correction, in red, blue (horizontally) or black (vertically) respectively when the two compared conditions are warm, cool or different. An asterisk besides an x-axis tick label shows that the corresponding mean feature is significantly different from 0 based on one sample *t*-tests with Holm-Bonferroni correction.

stimulation between the first and the second stimulation cycle. In contrast, maximum ratings of the intensity of the percept elicited by cool stimulation tended to decrease from the first to the second stimulation cycle, especially for stimuli delivered using a fixed surface. The ANOVA revealed significant main effect of temperature (cool vs. warm) on the change in rating between the first and second cycle of stimulation, but no main effect of stimulation surface

**Table 1. ANOVAs for the features of the intensity ratings.** Outcomes for the main effects and interactions from the repeated measures ANOVA performed for the average (two first rows) and dynamical (four last rows) features of the intensity ratings. Partial eta squared ($\eta_p^2$) are indicated for the effect sizes and the *p*- values smaller than the significance level of 0.05 are in bold.

|  |  | Temperature | | | Surface | | | Temperature*Surface | | |
|---|---|---|---|---|---|---|---|---|---|---|
|  |  | F | Prob>F | $\eta_p^2$ | F | Prob>F | $\eta_p^2$ | F | Prob>F | $\eta_p^2$ |
| Mean | Mean peak (a.u.) | 25.294 | **0.000** | 0.658 | 14.537 | **0.000** | 0.517 | 3.569 | **0.043** | 0.222 |
|  | Mean latency (s) | 46.763 | **0.000** | 0.781 | 8.040 | **0.013** | 0.378 | 6.724 | **0.005** | 0.350 |
| Early | $\delta$ Peak (%) | 8.606 | **0.011** | 0.395 | 0.237 | 0.790 | 0.017 | 1.016 | 0.377 | 0.075 |
|  | $\delta$ Latency (s) | 14.313 | **0.002** | 0.516 | 10.398 | **0.000** | 0.438 | 10.515 | **0.000** | 0.457 |
| Global | $\Delta$ Peak (%) | 6.058 | **0.028** | 0.314 | 10.107 | **0.001** | 0.428 | 0.558 | 0.579 | 0.043 |
|  | $\Delta$ Latency (s) | 3.291 | 0.092 | 0.200 | 0.009 | 0.951 | 0.001 | 1.360 | 0.275 | 0.098 |

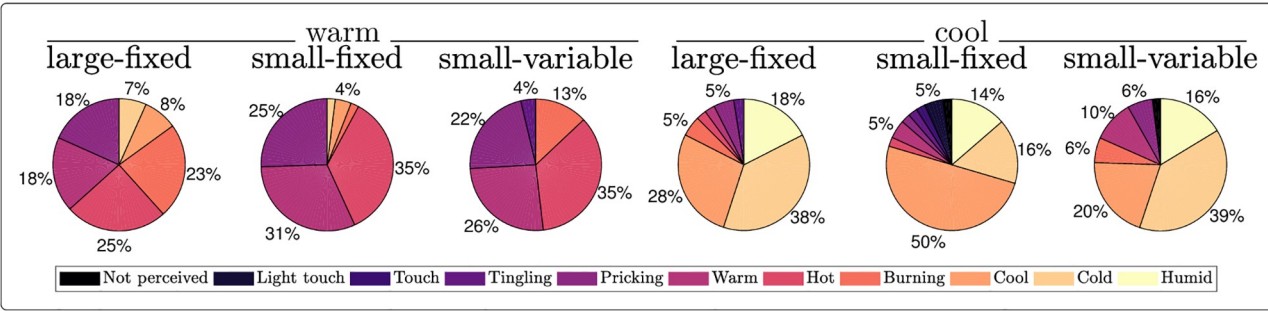

**Fig 4. Quality of the percepts.** Reported quality of the percepts from all the subjects and for each condition. The proportions below 3% are not indicated for clarity.

(fixed vs. variable), and no interaction between the two factors ('$\delta$ Peak' in Table 1). The latency of the maximum rating of intensity of perception also differed between the first and the second stimulation cycle. The ANOVA showed a main effect of temperature, a main effect of stimulation surface, and an interaction between the two factors ('$\delta$ Latency' in Table 1). Post-hoc comparisons showed that all rating latencies increased in the second compared to the first cycle, and that this increase was larger for cool compared to warm stimulation when the stimulation surface was fixed (Fig 3e).

**Global changes in perception occurring after 15 stimulation cycles.** The ANOVA comparing the changes in intensity ratings between the first and the last stimulation cycle revealed significant effects of both temperature and surface ('$\Delta$ Peak' in Table 1). Paired comparisons showed that, for cool stimulation, the global decrease in perception was stronger when stimulation was delivered using a fixed surface as compared to a variable surface (Fig 3f). The decrease in perception across the stimulation cycles was also greater for cool compared to warm stimulation, although the differences were not significant. The ANOVA comparing the changes in rating latencies did not show any significant effect ('$\Delta$ Latency' in Table 1).

The results of the ANOVA and post-hoc tests for the additional quantities that were compared are reported in S1 Table and S3 Fig. There was no significant effect of the temperature nor the surface on the minimum intensity ratings, and the effects on the amplitudes of the cyclic variations (rating ranges) were similar to the ones of the rating peaks reported above.

**Quality of the sensations elicited by heat and cool stimulation.** The descriptors chosen to describe sensation quality are shown in Fig 4. Periodic heat stimulation was most often described as 'warm', 'hot', 'burning' or 'pricking'. Periodic cool stimulation was most often described as 'cool', 'cold' or 'humid'.

## Experiment 2: Electroencephalographic recordings

**Frequency domain analysis.** The EEG frequency spectra at electrode FCz obtained during warm and cool stimulation using a fixed or variable surface are shown in Fig 5a. The periodicity was the largest on average at this electrode. For cool stimulation, a significant but small response was observed at the frequency of stimulation (0.2 Hz) when stimulation was delivered using a variable surface, and no response was observed when stimulation was delivered using a fixed surface. For warm stimulation, a markedly greater response was observed, both when delivered using a fixed surface and when delivered using a variable surface. The increase was significant at the frequency of stimulation (0.2 Hz) and the three following harmonics (0.4, 0.6 and 0.8 Hz). The ANOVA conducted on the noise-subtracted amplitude at 0.2 Hz revealed main effects of temperature and surface (Table 2). Post-hoc comparisons showed that the

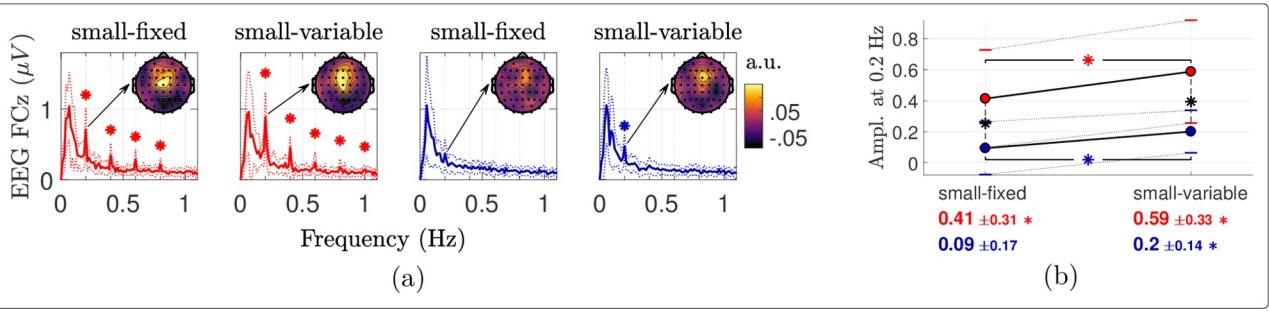

**Fig 5. EEG frequency analysis.** (a) Group-level average (± standard deviation in dotted) Fourier transforms of the EEG signals at electrode FCz. A star indicates significance of the noise-subtracted peak at $\{k \cdot 0.2\}_{k=1}^{5}$ Hz (t-tests against 0). The scalp maps show the distributions of the noise-subtracted amplitudes at 0.2 Hz. (b) Pairwise comparisons of the noise-subtracted EEG amplitudes at 0.2 Hz at electrode FCz. The red (resp. blue) dots show the mean amplitudes for the warm (resp. cool) stimulation, the standard deviations being indicated with horizontal bars. These means and standard deviations are also reported below the plots with the corresponding color. Each asterisk in the plot indicates a significant difference according to paired samples t-tests, in red, blue (horizontally) or black (vertically) respectively when the two compared conditions are warm, cool or different.

periodic EEG response was greater for warm vs. cool stimulation, and greater for stimulation using a variable vs. fixed surface (Fig 5b).

Group-level average scalp topographies of the noise-subtracted amplitudes at 0.2 Hz are shown in Fig 5a. In all conditions in which stimulation elicited a significant periodic EEG response, its topography was maximal over fronto-central electrodes, and symmetrically distributed over the two hemispheres.

**Time domain analysis.** The EEG signals averaged across all stimulation cycles are depicted in Fig 6a. At electrode FCz, the EEG response elicited by warm stimulation consisted in a positive wave peaking approximately 1 second after the peak of heat stimulation. For cool stimulation the response also appeared to consist of a positive wave, but its amplitude was much smaller than for warm stimulation, especially when stimulation was delivered using a fixed surface. The peak latency of the response to cool stimulation was also much shorter than the peak latency of the response to heat stimulation (Fig 6b; Table 2).

**Time-frequency analysis.** The STFT of the signals revealed a marked decrease of the components at 0.2 Hz in all conditions, which appeared to be greater for stimulation delivered using a fixed surface (Fig 7). This decrease was particularly strong and early for cool stimulation, as highlighted by the time course of the stimulus-evoked components at 0.2 Hz (bottom insets). The ANOVA performed on the AUC of these components (used as a measure of global response amplitude: the larger and the more persistent the elicited activity, the larger the AUC) confirmed that the elicited response was indeed stronger for the warm compared to the cool stimulation (Fig 8a and Table 2). Although not statistically significantly, using a variable surface induced a larger AUC than using a fixed surface. Besides, the noise-subtracted

**Table 2. ANOVAs for the features of the EEG signals.** Outcomes of the ANOVA performed to assess the main effects of the temperature and surface and their interaction on the EEG signals at FCz. Partial eta squared ($\eta_p^2$) are indicated for the effect sizes and the p- values smaller than the significance level of 0.05 are in bold.

| | Temperature | | | Surface | | | Temperature*Surface | | |
|---|---|---|---|---|---|---|---|---|---|
| | **F** | **Prob>F** | $\eta_p^2$ | **F** | **Prob>F** | $\eta_p^2$ | **F** | **Prob>F** | $\eta_p^2$ |
| Ampl. at 0.2 Hz | 23.690 | **0.000** | 0.629 | 10.905 | **0.005** | 0.438 | 0.894 | 0.360 | 0.060 |
| Mean lat. (s) | 25.926 | **0.000** | 0.649 | 0.042 | 0.841 | 0.003 | 0.271 | 0.611 | 0.019 |
| AUC at 0.2 Hz (a.u.) | 27.969 | **0.000** | 0.666 | 8.982 | **0.010** | 0.391 | 0.119 | 0.735 | 0.008 |

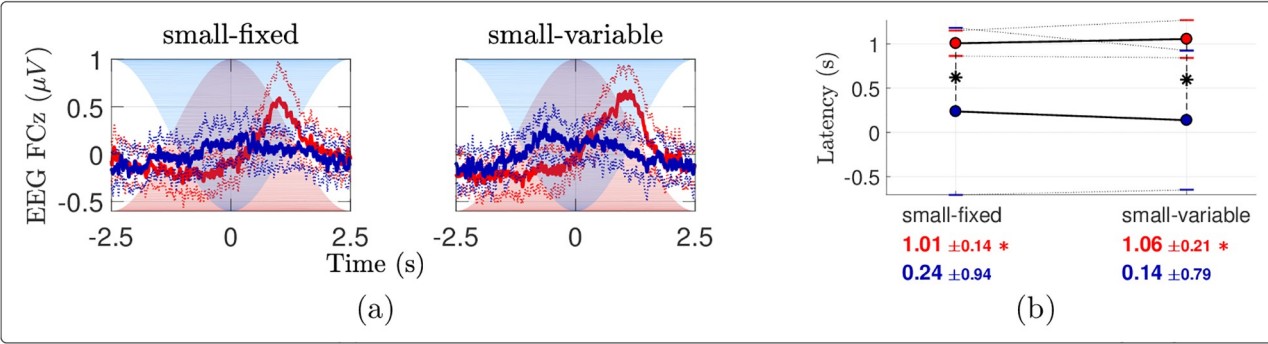

**Fig 6. EEG averaged across stimulation cycles.** (a) Group-level average (± standard deviation in dotted) time courses of the EEG signals at electrode FCz averaged across stimulation cycles, in (light) red and (dark) blue for the warm and cool conditions respectively. The stimulation temperatures are shaded. (b) Pairwise comparisons of the latencies of the EEG peaks compared to the temperature peaks. The red (resp. blue) dots show the mean latencies for the warm (resp. cool) stimulation, the standard deviations being indicated with horizontal bars. These means and standard deviations are also reported below the plots with the corresponding color. Each asterisk in the plot indicates a significant difference across temperature according to paired samples *t*-tests. An asterisk besides an x-axis tick label shows that the corresponding mean latency is significantly different from 0 based on one sample *t*-tests with Holm-Bonferroni correction.

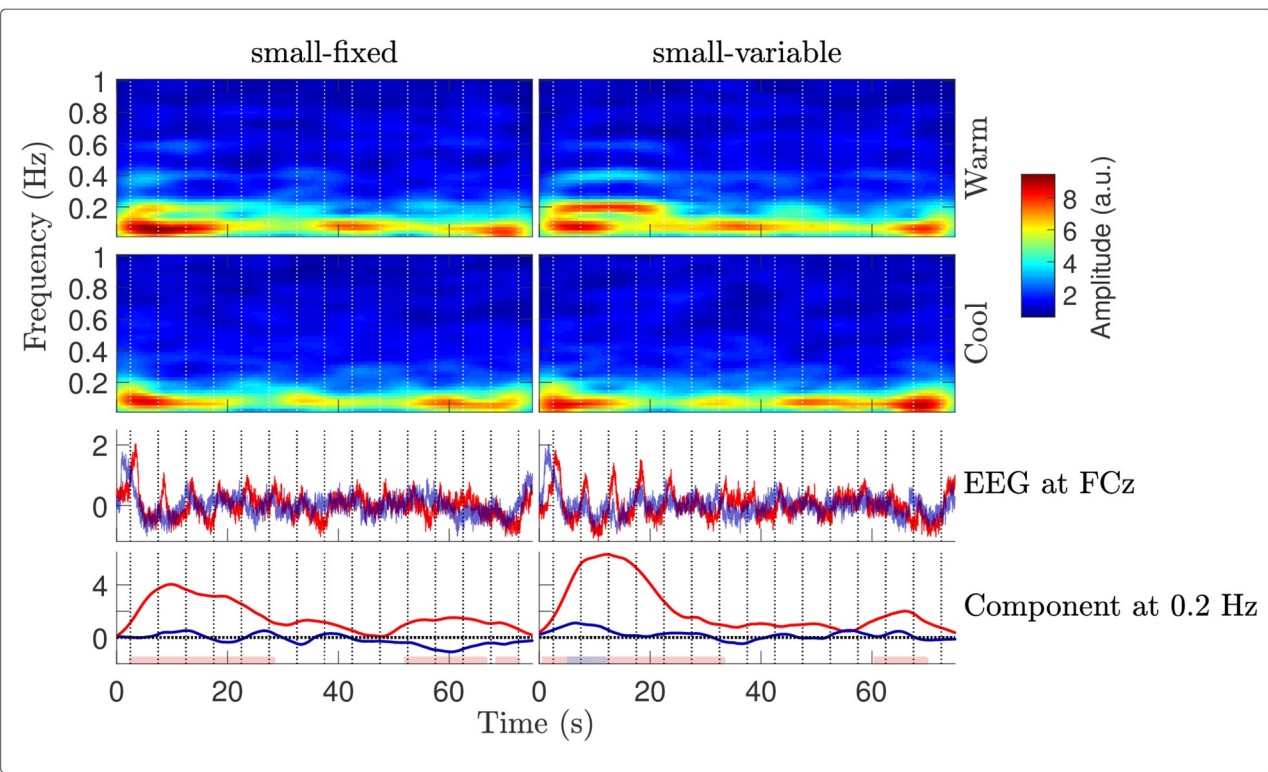

**Fig 7. EEG time-frequency analysis.** Amplitude of the STFT of the EEG signals at FCz, averaged across subjects. The first (resp. second) row of bottom insets displays the grand average time courses (resp. the stimulus-evoked EEG components at 0.2 Hz, estimated as the sum of the noise-subtracted STFT amplitudes at $\{k \cdot 0.2\}_{k=1}^{5}$ Hz), in red and blue for the warm and cool conditions. The temperature peaks in each cycle are indicated with vertical dotted lines. The shaded horizontal bars at the bottom of the figure show, for each of the four conditions, the significant time clusters where the depicted noise-subtracted amplitude at 0.2 Hz is significantly greater than 0 (*t*-tests against 0).

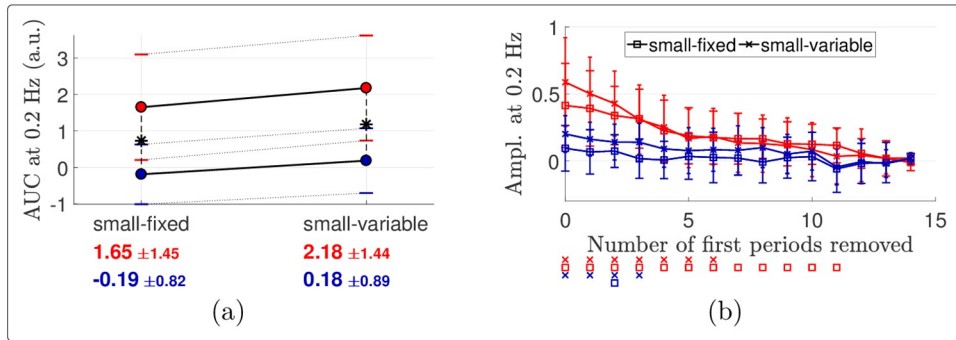

**Fig 8. Habituation of the EEG components at 0.2 Hz.** (a) Effects of the stimulation temperature and surface on the AUC of the stimulus-evoked EEG component at 0.2 Hz at FCz, estimated as the sum of the noise-subtracted STFT amplitudes at $\{k \cdot 0.2\}_{k=1}^{5}$ Hz. The red (resp. blue) dots show the mean amplitudes for the warm (resp. cool) stimulation, the standard deviations being indicated with horizontal bars. These means and standard deviations are also reported below the plots with the corresponding color. Each asterisk in the plot indicates a significant difference across temperature according to paired samples $t$-tests. (b) Noise-subtracted amplitudes of the Fourier transforms at 0.2 Hz at FCz as a function of the number of periods removed at the beginning of the EEG signals. The error bars show the standard deviations across subjects, with larger caps when the surface is fixed than when it is variable. For each number of periods removed along the $x$-axis, a marker drawn below the plot indicates that the noise-subtracted amplitude in the associated condition (with the same color and marker) and for this abscissa is significantly different from 0, according to one-sample $t$-tests with Holm-Bonferroni correction.

amplitudes of the FT at 0.2 Hz (Fig 8b) indicated that there was a significant periodic EEG response at FCz until 7 (or 12) periods were removed in the warm condition with a variable (or fixed, respectively) surface. Also, the periodicity remained significant for the cool condition with a variable surface when up to 3 cycles were removed, showing that a stimulus-evoked response persisted for at least a few cycles.

**Modulation of ongoing oscillations.**   Warm stimulation induced a periodic modulation (reduction) of the power of alpha- and beta-band oscillations, associated with a scalp topography maximal over centro-parietal areas contralateral to the stimulated arm (especially at C3), as illustrated in panels (a) and (b) of Figs 9 and 10. Like for the baseband signal, the cool stimulation elicited similar modulations, but with smaller magnitudes and shorter latencies. All responses tended to decrease along the stimulation cycles, especially with a fixed stimulated surface (see Figs 9c, 9e, 10c and 10e). There was no consistent and significant modulation of theta and gamma power. The results of the analysis of these envelopes are therefore provided as Supplementary Material, see S4 and S5 Figs.

## Discussion

The objective of this study was to compare the perception and EEG responses elicited by long-lasting warm and cool stimuli frequency-tagged by slowly and periodically varying their intensity over time at a frequency of 0.2 Hz, and applied either to the same patch of skin or to a varying patch of skin along the stimulation periods.

### Links between perception and EEG responses

Although the perceptual ratings and EEG recordings have different natures and signal-to-noise ratios, some of their properties can be linked. On average, the magnitudes of both the periodic percept (Fig 2b) and the EEG response at 0.2 Hz (Figs 5b and 8a) were larger for warm than cool stimulation, and also larger when the surface was variable compared to fixed in the cool case. In the warm case, only the stimulus-evoked EEG response was increased by

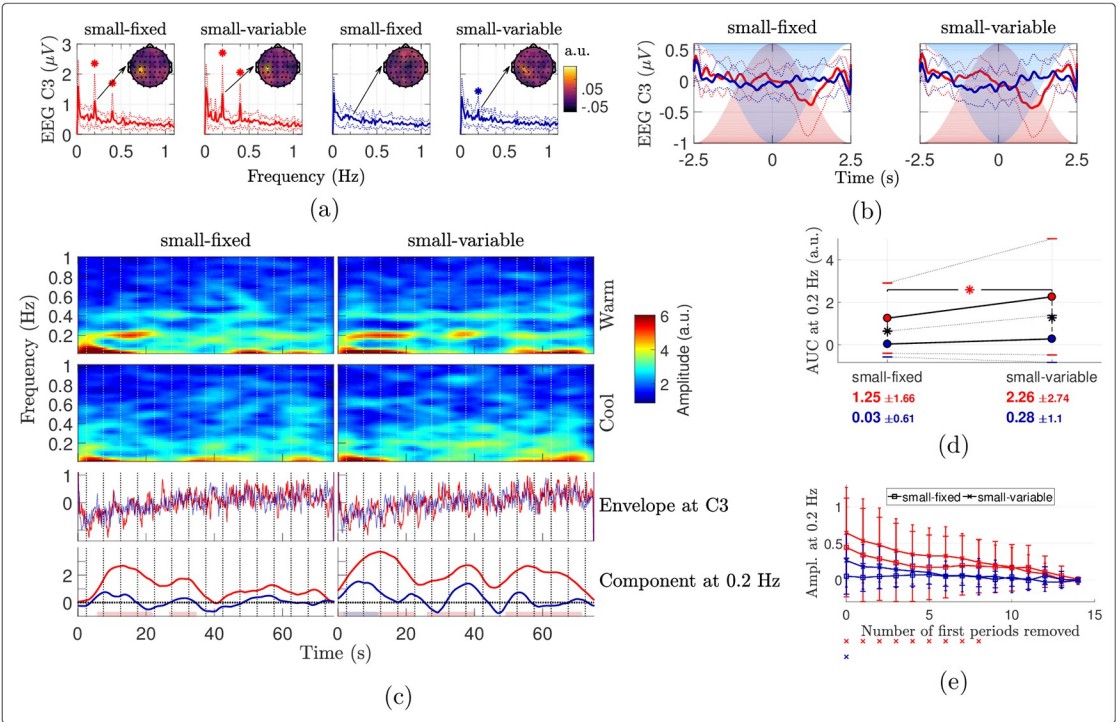

**Fig 9. Signal envelopes in the alpha frequency band (8-12 Hz).** (a) Fourier transforms of the envelopes averaged over the epochs (grand mean ± standard deviation) at electrode C3. A star indicates significance of the noise-subtracted peak at $\{k \cdot 0.2\}_{k=1}^{5}$ Hz (*t*-tests against 0). The scalp maps show the distributions of the noise-subtracted amplitudes at 0.2 Hz. (b) Envelopes averaged across stimulation periods (grand mean ± standard deviation). (c) Grand mean STFT of the average signals at C3 and (d) pairwise comparisons of the AUC of the stimulus-evoked envelope components at 0.2 Hz. The red (resp. blue) dots show the mean AUCs for the warm (resp. cool) stimulation, the standard deviations being indicated with horizontal bars. Each asterisk in the plot indicates a significant difference according to paired samples *t*-tests, in red, blue or black respectively when the two compared conditions are warm, cool or different. (e) Noise-subtracted amplitudes of the Fourier transforms at 0.2 Hz at C3 as a function of the number of periods removed at the beginning of the envelopes. The error bars show the standard deviations across subjects, with larger caps when the surface is fixed than when it is variable. For each number of periods removed along the *x*-axis, a marker drawn below the plot indicates that the noise-subtracted amplitude in the associated condition (with the same color and marker) and for this abscissa is significantly different from 0, according to one-sample *t*-tests with Holm-Bonferroni correction.

using a variable surface. Yet, the significant main effect of the surface on the global rating changes (without significant temperature—surface interaction, see Table 1) indicates that the warm perception also decreased more with a fixed surface than with a variable one. Then, the EEG and rating latencies were smaller for cool than warm stimuli (Figs 2c and 6b). In terms of perception, varying the stimulated skin surface increased the latencies for cool stimuli while such difference was not observed with the EEG. However, the EEG latency for cool stimuli delivered on a fixed surface was not very reliable since the periodic response was not significant. The intensity ratings and EEG recordings can be less accurately compared in terms of dynamics, as the maximum rated intensity and its latency could be extracted during each stimulation cycle whereas a clear stimulus-evoked EEG response could not be identified for each cycle. Nevertheless, the global and early changes in intensity rating (Fig 3) show that habituation of the perception was stronger and earlier for the cool stimuli, especially with a fixed stimulation area. Likewise, the amplitude at 0.2 Hz became non-significant sooner for the cool than the warm stimulation as the first cycles were removed from the EEG signals (Fig 8b), with no significant periodic response at all for a cool fixed stimulation. All these large similarities

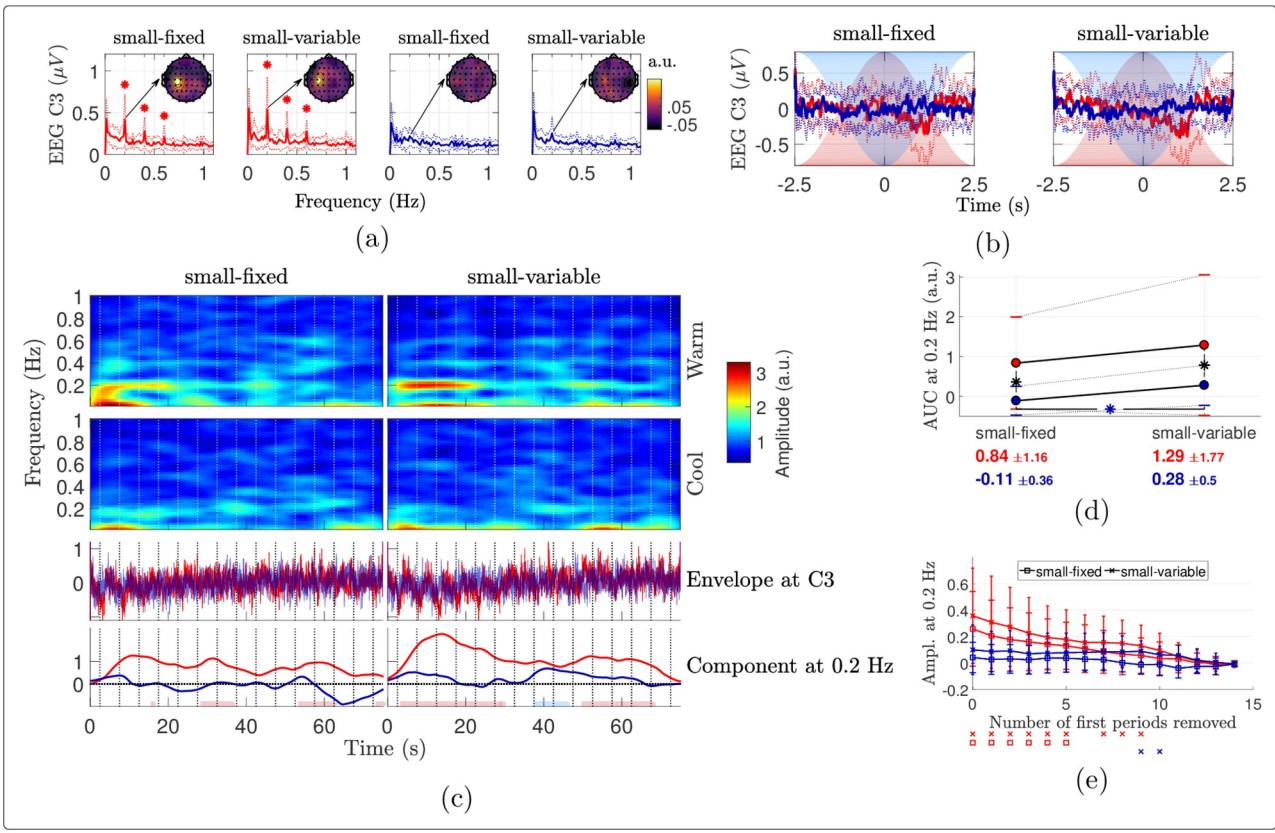

**Fig 10. Signal envelopes in the beta frequency band (12-30 Hz).** (a) Fourier transforms of the envelopes averaged over the epochs (grand mean ± standard deviation) at electrode C3. A star indicates significance of the noise-subtracted peak at $\{k \cdot 0.2\}_{k=1}^{5}$ Hz (*t*-tests against 0). The scalp maps show the distributions of the noise-subtracted amplitudes at 0.2 Hz. (b) Envelopes averaged across stimulation periods (grand mean ± standard deviation). (c) Grand mean STFT of the average signals at C3 and (d) pairwise comparisons of the AUC of the stimulus-evoked envelope components at 0.2 Hz. The red (resp. blue) dots show the mean AUCs for the warm (resp. cool) stimulation, the standard deviations being indicated with horizontal bars. Each asterisk in the plot indicates a significant difference according to paired samples *t*-tests, in red, blue or black respectively when the two compared conditions are warm, cool or different. (e) Noise-subtracted amplitudes of the Fourier transforms at 0.2 Hz at C3 as a function of the number of periods removed at the beginning of the envelopes. The error bars show the standard deviations across subjects, with larger caps when the surface is fixed than when it is variable. For each number of periods removed along the *x*-axis, a marker drawn below the plot indicates that the noise-subtracted amplitude in the associated condition (with the same color and marker) and for this abscissa is significantly different from 0, according to one-sample *t*-tests with Holm-Bonferroni correction.

between the perceptual outcomes and EEG features hence suggest that the EEG responses were mostly correlated with perception.

## Consistency with laser stimulation

In previous studies using laser heat stimulation, we showed that slowly and sinusoidally heating the skin between baseline and 50°C at a frequency of 0.2 Hz elicits a periodic EEG response mainly driven by the activation of unmyelinated C-fibers [29]. Indeed, selectively blocking the conduction of myelinated fibers did not alter the elicited SSRs. In the present study, periodic contact heat stimulation at 0.2 Hz elicited a similar SSR, maximal at the scalp vertex and symmetrically distributed over the two hemispheres. Furthermore, such as in Colon et al., periodic contact heat stimulation at 0.2 Hz elicited a periodic modulation of ongoing oscillations, with a scalp topography maximal over parietal regions contralateral to the stimulated limb.

## EEG responses evoked by cool stimulation

Innocuous periodic cooling of the skin at the same frequency also elicited a periodic EEG response. However, as compared to periodic heat stimulation, the magnitude of cool-evoked SSRs was markedly lower. This suggests that, at 0.2 Hz, the periodic activity generated by the activation of cool-sensitive afferents using contact cooling of the skin is not as strong as the periodic activity generated by the activation of heat-sensitive afferents. Besides, the cool SSRs could also be influenced by the periodic suppression of spontaneous tonic activity within warm-sensitive afferents [5, 56].

## Habituation of perception and EEG responses

Both heat- and cool-evoked responses tended to attenuate along the stimulation cycles. Possible explanations for this response attenuation are receptor fatigue or adaptation at peripheral level and/or habituation processes or alternative phenomena occurring at the level of the central nervous system [35, 38]. This response attenuation could explain why both heat-evoked and cool-evoked SSRs tend to be of smaller magnitude than the SSRs typically elicited by other types of stimuli, such as vibrotactile, visual or auditory stimuli [30, 57, 58]. The finding that attenuation of the EEG response and perception over time is stronger for cool as compared to heat stimulation could be explained by (1) peripheral mechanisms differentially affecting the responsiveness of heat vs. cool-sensitive afferents and/or (2) central mechanisms specific to the painful nature of the heat stimuli compared to the cool ones. However, further supporting a contribution of peripheral mechanisms is the observation that the response attenuation was less pronounced when the stimulated skin area was varied along the stimulation cycles, i.e. when different free nerve endings were exposed to heat or cold across the stimulation cycles [35]. Notably, when Colon et al. recorded heat-evoked SSRs using infrared laser stimulation, the laser beam was displaced between each stimulation cycle such that stimulation was never repeated at the same skin area over the entire duration of the periodic stimulus [29]. Using such stimuli, no habituation of perception or SSRs was observed, suggesting a limited contribution of central habituation. Therefore, the reduced steadiness of the EEG responses when the warm stimulation is applied on a fixed skin location further suggests that rather quickly-adapting fibers were, at least partly, responsible for the EEG responses elicited by the first cycles of warm stimulation [6, 59].

## Differences in latency of the responses

Another marked difference between the responses elicited by periodic heat and cool stimulation was a difference in latency. Regarding the time course of perception, the periodic variations of the intensity of heat perception were delayed by approximately 0.8 seconds as compared to the periodic variations of the intensity of cool perception. A similar delay of approximately 0.8 seconds was observed when comparing the time course of heat- and cool-evoked SSRs. These differences in response latency should be interpreted with caution, as they could be explained by at least three factors. First, these differences could be related to differences in peripheral conduction times, considering that the responses elicited by sinusoidal heating of the skin at 0.2 Hz are predominantly related to the activation of slowly-conducting unmyelinated C-fibers [29], whereas cool stimuli might be mostly conveyed by faster-conducting thinly-myelinated A -fibers [1, 5]. However, the observed latency differences could also be explained by differences in timings of peak discharge frequency and in relative activation thresholds. Single-unit recordings in monkeys have shown that C fiber thermonociceptors can be either quickly adapting (QC) or slowly adapting (SC) [4, 6]. Exposed to a step increase of temperature from baseline to 49˚C maintained during 30 seconds, QCs respond vividly at the

onset of the stimulus with a peak discharge occurring approximately 0.4 s after stimulation onset, and then rapidly adapt. Conversely, SCs respond more gradually to the step increase in skin temperature, with a peak discharge approximately 2 seconds after stimulation onset, and then tend to maintain a tonic level of activity during the entire stimulus duration. Besides, the thermal activation threshold of QCs and SCs have been shown to be around 41˚C and 46˚C respectively [6]. The response properties of cool-sensitive free nerve endings have not been characterized as extensively [1, 5]. Humans are able to detect transient decreases in skin temperature of as little as 0.2-0.5˚C [60], and to respond sharply at the onset of the stimulus [61]. Therefore, when cooling the skin progressively from baseline to 14˚C, activation of cool-sensitive afferents may be expected to occur earlier than the activation of QC and/or SC fibers when the skin was progressively heated to 48˚C, both because thermal activation threshold was reached earlier in the stimulation cycle for cool stimulation as compared to heat stimulation, and because cool-sensitive afferents may be expected to reach peak discharge rate earlier than C fibers [6, 61].

## Specificity of cool perception

Interestingly, the latency of the peak of cool perception across stimulation cycles was markedly affected by varying the stimulated skin area, with an important increase of latency occurring after the first stimulation cycle when the surface was kept fixed. This phenomenon could be explained by activity-dependent slowing [7] and/or peripheral adaptation of cool-sensitive receptors, leading to an increase in their threshold [36].

## Modulation of ongoing oscillations

The temporal structure of periodic stimuli nicely allows to assess the presence of modulations of ongoing oscillations and to study their time dynamics. Our analyses showed that the periodic warm stimuli negatively modulated ongoing alpha and beta oscillations in contralateral centro-parietal areas. The periodic cool stimuli induced similar reductions of ongoing oscillations, with however smaller magnitudes and shorter latencies. These modulations closely resembled the baseband SSRs in terms of latency and reversed shape. These features are compatible with previous studies suggesting that tonic (cold or heat) pain could be accompanied by a reduction of alpha and beta power maximal over contralateral sensorimotor areas [23, 24, 26, 27, 62]. Such reduction of alpha (and sometimes beta) activity is also similarly observed following transient painful stimuli [26, 63, 64]. In this case, it is commonly referred to as alpha event-related desynchronization ($\alpha$-ERD) [65]. The $\alpha$-ERD is typically observed over the contralateral sensorimotor cortex and in occipital areas [63]. This reduction of alpha (and beta) power, probably originating from sensorimotor areas, is therefore common to both tonic and brief painful stimuli, and it likely encodes stimulus intensity [24, 66]. Coherently with this hypothesis, non-painful cool stimulation also significantly reduced alpha and beta powers in our study, with a smaller magnitude correlating with the smaller perceived intensity, with shorter latencies and with stronger habituation, as for the baseband responses. Habituation also tended to be enhanced when the stimulated skin surface was kept fixed.

Besides, we did not observed a consistent and significant modulation of gamma power, although an enhancement of gamma power during tonic painful stimulation had been reported in prefrontal regions [24] or widespread across the scalp [26], and could be specific to tonic pain perception [63]. The absence of a clear gamma modulation in our study could be explained by (1) the limited duration of our stimuli (75 seconds), (2) the overall habituation of the responses and/or (3) the large difference between the stimulation period (5 seconds) and the short time scales on which gamma oscillations evolve.

### Relations with imaging studies

The contralateral centro-parietal modulations of alpha and beta oscillations described above likely originate, at least partly, from the contralateral sensorimotor cortex [24]. Besides, the scalp topography of the EEG responses at 0.2 Hz, maximal over fronto-central electrodes and symmetrically distributed over the two hemispheres, could rather result from brain activity originating from the anterior cingulate cortex and/or the operculo-insular cortices [31]. Our findings are therefore compatible with functional neuroimaging studies showing that experimental tonic pain is accompanied by an increase in cerebral blood flow (CBF) in a large network of brain areas including the cingulate, primary and secondary somatosensory, prefrontal and insular cortices [18–20, 67, 68]. The fact that the magnitude of the phase-locked stimulus-evoked responses at 0.2 Hz and the modulations of ongoing oscillations had different scalp topographies further suggests that the widespread increases in CBF could be linked to distinct EEG features, such as a reduction of alpha-band oscillations in somatosensory areas and a phase-locked response from the insular and/or cingulate cortices. Further studies will be needed to functionally characterize all the components of the EEG responses and to better relate them to the brain structures highlighted in fMRI studies.

### Scope of the study and future works

This study compared long-lasting warm and cool stimulation. The presented results naturally depend on the selected frequency of stimulation, the waveform shape and the temperature delta, and varying these parameters may lead to different outcomes, as different populations of afferent fibers could be selectively activated and/or deactivated [4]. In this paper, the stimulus-induced activations of afferents were mainly discussed since literature results suggest that they should mostly determine the elicited responses [5]. However, these responses can also be affected by other stimulus-induced changes within the afferents, as studies have shown that some thermonociceptors (e.g. warm-sensitive C fibers) may maintain a tonic activity at rest, whose suppression by a (e.g. cool) stimulus could contribute to the elicited (e.g. cool) sensations and brain responses [5, 56].

Besides, while the stimulus duration (75 seconds) was limited compared to usual constant tonic thermal stimulation, the proposed approach enables studying the responses elicited by particular stimulation waveforms (e.g. lasting 5 seconds in the present work). Finally, given the habituation observed in our recordings and the relatively weak SSR, future studies will improve the functional characterization of the elicited brain responses across all the scalp channels, including the modulations of ongoing oscillations, for instance by testing painful and non-painful heat and cold stimuli and comparing the timing and localization of the obtained responses [64].

### Conclusion

In summary, both sinusoidal contact heat stimulation and sinusoidal contact cold stimulation at 0.2 Hz elicit a sensation whose intensity varies periodically over time. Furthermore, both stimuli elicit a periodic EEG response at 0.2 Hz and its harmonics, although the EEG response elicited by cool stimulation is of much lower magnitude than the EEG response elicited by warm stimulation. The latencies of the perception and EEG responses elicited by cool stimulation were, on average, shorter than those elicited by warm stimulation. This latency difference was most pronounced during the first cycle of stimulation. Both the perception and the EEG responses to warm and cool stimulation tended to habituate over time. This habituation was stronger for cool stimulation as compared to warm stimulation. Response habituation was less

pronounced when stimuli were delivered using a variable surface as compared to a fixed surface.

## Supporting information

**S1 Fig. Individual rating peaks and troughs across cycles.** The amplitude of the rating peaks as a function of the cycle index is depicted in blue, for the warm (top row) and cool (bottom row) stimulation. Each curve is normalized by the first peak amplitude. The grand average is in black. The pink curves indicate the minimum rating amplitudes reported between the corresponding peaks (the dotted black lines being their averages).
(PDF)

**S2 Fig. Individual latencies of the rating peaks across cycles.** Latency between the temperature and rating peaks as a function of the cycle index, for the warm (top row) and cool (bottom row) stimulation. The grand average is in black.
(PDF)

**S3 Fig. Post-hoc comparisons of the amplitudes of the cyclic variations in intensity ratings.** Pairwise comparisons for the effects of the temperature and surface (a) on the rating range averaged across cycles, (b) on the early difference of rating range and (c) on the global difference of rating range across cycles. The dots colored in red (resp. blue) show the mean features in the warm (resp. cool) conditions. These means (with the standard deviations) are also indicated below the plots with the same color. Each asterisk in the plot indicates a significant difference according to paired sample $t$-tests with Holm-Bonferroni correction, in red, blue or black respectively when the two compared conditions are warm, cool or different. In (b,c), an asterisk besides an x-axis tick label shows that the corresponding mean feature is significantly different from 0 based on one sample $t$-tests with Holm-Bonferroni correction. These comparisons of the rating ranges are similar to the ones of the rating peaks presented in the paper.
(PDF)

**S4 Fig. Signal envelopes in the theta frequency band (4-8 Hz).** (a) Fourier transforms of the envelopes averaged over the epochs (grand mean ± standard deviation) at electrode CP1. A star indicates significance of the noise-subtracted peak at $\{k \cdot 0.2\}_{k=1}^{5}$ Hz ($t$-tests against 0). The scalp maps show the distributions of the noise-subtracted amplitudes at 0.2 Hz. (b) Envelopes averaged across stimulation periods (grand mean ± standard deviation). (c) Grand mean STFT of the average signals at CP1 and (d) pairwise comparisons of the AUC of the stimulus-evoked envelope components at 0.2 Hz. The red (resp. blue) dots show the mean AUCs for the warm (resp. cool) stimulation, the standard deviations being indicated with horizontal bars. Each asterisk in the plot indicates a significant difference according to paired samples $t$-tests, in red, blue or black respectively when the two compared conditions are warm, cool or different. (e) Noise-subtracted amplitudes of the Fourier transforms at 0.2 Hz at CP1 as a function of the number of periods removed at the beginning of the envelopes. The error bars show the standard deviations across subjects, with larger caps when the surface is fixed than when it is variable. For each number of periods removed along the $x$-axis, a marker drawn below the plot indicates that the noise-subtracted amplitude in the associated condition (with the same color and marker) and for this abscissa is significantly different from 0, according to one-sample $t$-tests with Holm-Bonferroni correction.
(PDF)

**S5 Fig. Signal envelopes in the gamma frequency band (30-50 Hz).** (a) Fourier transforms of the envelopes averaged over the epochs (grand mean ± standard deviation) at electrode C1.

A star indicates significance of the noise-subtracted peak at $\{k \cdot 0.2\}_{k=1}^{5}$ Hz (*t*-tests against 0). The scalp maps show the distributions of the noise-subtracted amplitudes at 0.2 Hz. (b) Envelopes averaged across epochs that were first low-pass filtered below 2 Hz to be able to identify the nature of the periodic modulations and then averaged across stimulation periods (grand mean ± standard deviation). (c) Grand mean STFT of the average signals at C1 and (d) pairwise comparisons of the AUC of the stimulus-evoked envelope components at 0.2 Hz. The red (resp. blue) dots show the mean AUCs for the warm (resp. cool) stimulation, the standard deviations being indicated with horizontal bars. Each asterisk in the plot indicates a significant difference according to paired samples *t*-tests, in red, blue or black respectively when the two compared conditions are warm, cool or different. (e) Noise-subtracted amplitudes of the Fourier transforms at 0.2 Hz at C1 as a function of the number of periods removed at the beginning of the envelopes. The error bars show the standard deviations across subjects, with larger caps when the surface is fixed than when it is variable. For each number of periods removed along the *x*-axis, a marker drawn below the plot indicates that the noise-subtracted amplitude in the associated condition (with the same color and marker) and for this abscissa is significantly different from 0, according to one-sample *t*-tests with Holm-Bonferroni correction.
(PDF)

**S1 Table. ANOVAs for all features of the intensity ratings.** Outcomes for the main effects and interactions from the repeated measures ANOVA performed for all the average (four first rows) and dynamical (eight last rows) features of the intensity ratings. Partial eta squared ($\eta_P^2$) are indicated for the effect sizes and the *p*- values smaller than the significance level of 0.05 are in bold. The results regarding the rating troughs and ranges are not reported in the paper as there was no significant effect for the rating troughs, and the effects for the rating ranges were similar to the ones for the rating peaks (presented in the paper).
(PDF)

## Author Contributions

**Conceptualization:** Dounia Mulders, Nicolas Lejeune, André Mouraux.

**Data curation:** Dounia Mulders.

**Formal analysis:** Dounia Mulders, Cyril de Bodt.

**Funding acquisition:** Dounia Mulders, Michel Verleysen, André Mouraux.

**Investigation:** Dounia Mulders, Nicolas Lejeune.

**Methodology:** Dounia Mulders, Cyril de Bodt, André Mouraux.

**Resources:** André Mouraux.

**Software:** Dounia Mulders, André Mouraux.

**Supervision:** Michel Verleysen, André Mouraux.

**Visualization:** Dounia Mulders, André Mouraux.

**Writing – original draft:** Dounia Mulders, André Mouraux.

**Writing – review & editing:** Dounia Mulders, Cyril de Bodt, Arthur Courtin, Giulia Liberati, Michel Verleysen, André Mouraux.

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
