## [Decision Letter · Decision Letter 0]

16 Dec 2019

PONE-D-19-20397

Dynamics of the perception and EEG signals triggered by tonic warm and cool stimulation

PLOS ONE

Dear Ms. Mulders,

Thank you for submitting your manuscript to PLOS ONE. After careful consideration, we feel that it has merit but does not fully meet PLOS ONE’s publication criteria as it currently stands. Therefore, we invite you to submit a revised version of the manuscript that addresses the points raised during the review process.

The manuscript has been assessed by four reviewers; their comments are available below.

The reviewers find the work of relevance but have raised a number of items that need attention in a revision, the reviewers request clarification regarding aspects of the study design and data analysis, and they have recommendations for the statistical analyses, interpretation of the findings as well as the presentation in figures and main text.

Could you please revise the manuscript to carefully address the concerns raised by the reviewers?

We would appreciate receiving your revised manuscript by Jan 28 2020 11:59PM. To enhance the reproducibility of your results, we recommend that if applicable you deposit your laboratory protocols in protocols.io, where a protocol can be assigned its own identifier (DOI) such that it can be cited independently in the future. For instructions see: http://journals.plos.org/plosone/s/submission-guidelines#loc-laboratory-protocols

We look forward to receiving your revised manuscript.

Kind regards,

Iratxe Puebla

Senior Managing Editor, PLOS ONE

Journal Requirements:

**When submitting your revision, we need you to address these additional requirements:**

**Please ensure that your manuscript meets PLOS ONE's style requirements, including those for file naming. The PLOS ONE style templates can be found at http://www.plosone.org/attachments/PLOSOne_formatting_sample_main_body.pdf and http://www.plosone.org/attachments/PLOSOne_formatting_sample_title_authors_affiliations.pdf**

Reviewers' comments:

Reviewer's Responses to Questions

**Comments to the Author**

1. Is the manuscript technically sound, and do the data support the conclusions?

Reviewer #1: Yes

Reviewer #2: Yes

Reviewer #3: Yes

Reviewer #4: Yes

2. Has the statistical analysis been performed appropriately and rigorously? 

Reviewer #1: Yes

Reviewer #2: Yes

Reviewer #3: I Don't Know

Reviewer #4: Yes

3. Have the authors made all data underlying the findings in their manuscript fully available?

Reviewer #1: Yes

Reviewer #2: Yes

Reviewer #3: No

Reviewer #4: Yes

4. Is the manuscript presented in an intelligible fashion and written in standard English?

Reviewer #1: Yes

Reviewer #2: Yes

Reviewer #3: Yes

Reviewer #4: Yes

5. Review Comments to the Author

Reviewer #1: The authors used oscillating warm and cool stimuli to investigate the dynamics of thermal perception and the steady-state EEG responses to thermal stimulation. The research and analyses were performed to a high standard. Moreover, the manuscript is well written, cogent, and discusses the results appropriately, with reference to relevant advances in the field. I commend the authors for their work and believe it warrants publication in PLoS ONE. However, there are a few issues I would like the authors to address first.

1) You chose the frequency and amplitude of the warm and cool temperature oscillations based on the optimal parameters for stimulating heat-sensitive C-fiber afferents (see p. 6). It is perhaps not surprising, then, that you obtained stronger steady-state EEG responses to your oscillating warm stimuli than your oscillating cool stimuli. I would like to see you discuss this issue in your manuscript. Do you think you might have obtained different results if you had optimized your stimulation parameters for activating cool-sensitive afferents, instead?

2) You did not specify the inter-stimulus interval in the first experiment. Was it the same as in Experiment 2 (i.e., 10-20 s)?

3) Your visual analog scale was labeled with “lowest reported intensity” for the minimum and “highest reported intensity” for the maximum. Did “reported intensity” refer to previous reports that participants made before the main experiment, or to the lowest and highest intensity reports that they could imagine? And did you instruct participants to report pain intensity or the intensity of thermal sensation?

4) On p. 8, you describe your process of normalizing the VAS ratings based on the minimum rating (“0”) and maximum rating (“1”) given in the first stimulation cycle. My reading of that passage was that you normalized ratings in this way for each individual stimulus. I would thus have expected that the maximum rating in the first cycle would always be one, but that is not the case in Fig. 3 (see top left). Could you please clarify your normalization procedure?

5) On p. 10, you wrote that you rejected EEG epochs with large artifacts in them, by visual inspection. Given that your epochs were 75 s long, I would think that all of them contained at least some blink artifacts. Could you explain how you dealt with those artifacts, other than by rejecting the entire epoch?

6) For your analysis of the steady-state EEG responses in the frequency domain, you obtained noise-subtracted spectra by subtracting the average signal amplitude at 8 neighboring frequencies from the average amplitude at the frequency of interest (FOI). Could you please specify how far those neighboring frequencies were from the FOI (i.e., what were the frequency steps of your Fourier transform)?

7) Why did you focus your EEG analyses on the signal from electrode FCz? Did you choose that electrode a priori, or was your decision based on where the steady-state response was maximal?

8) Your time-frequency analysis used a continuous wavelet transform with a complex Morlet wavelet. This means that higher frequencies had better temporal resolution, but worse spectral resolution, compared with lower frequencies. After performing that transform, you computed noise-subtracted time-frequency maps by subtracting the signal amplitude of neighboring frequencies from each frequency amplitude over time. Was this process complicated by the fact that the neighboring frequencies had somewhat different temporal resolutions? Moreover, did the distance between neighboring frequencies increase as the FOI increased, because of the lower spectral resolution at higher frequencies?

9) I was a bit surprised that you didn’t look at habituation of the steady-state response in your time-frequency data. Instead, you computed multiple Fourier transforms with an increasing number of periods removed from the beginning of the stimulus. Could you explain why you chose that approach?

10) Your finding that periodic warm and/or cool stimulation modulated ongoing oscillations in frontal gamma oscillations and in contralateral parietal alpha and beta oscillations is interesting, given the roles of those oscillations in pain and sensorimotor processes. I think it would be useful to discuss those findings further, and perhaps present the relevant figures in the main text, if there is space.

Additionally, I have a few minor suggestions to further improve the clarity and readability of the manuscript:

11) In your abstract, you talk about "displacement of the stimulated skin surface". Initially, I thought you meant that you had displaced the skin itself. You might consider rewording this, e.g., "displacement of thermal stimulation on the skin surface".

12) Your repeated use of the abbreviation “resp.” (“respectively”) in parentheses is sometimes confusing. You might consider rephrasing relevant sentences, e.g., “Since the evolution of the intensity of a long-lasting sensation is typically nonlinear, with an early strong decrease (or increase) when habituation (or sensitization, respectively) occurs…”

13) The data markers you used in Fig. 3 to denote the different stimulation conditions (top right) are a bit small and difficult to distinguish from each other. You could perhaps use different and/or larger markers for better readability.

14) For readers who are not very familiar with steady-state evoked responses in EEG, it might be helpful to explain why you looked at the harmonics of 0.2 Hz, in addition to the fundamental frequency.

15) “Significativity” should be “significance” (see figure captions).

Reviewer #2: This is an elegant paper demonstrating both perception and EEG signatures for sinusoidal heat and cold stimuli. The authors properly discuss the literature on the topic in the introduction. The methods seem complete and data analysis is properly executed. I only have minor questions, suggestions.

1) There is no effort of linking EEG properties to perceptual outcomes. Even a discussion of the topic would be important.

2) The introduction expounds, rather elegantly, about fiber subtypes and their responses to different stimuli. This topic is not addressed at the end of the paper. The discussion should include some linkages between observed results and afferent fiber responses in this specific experimental setting.

3) The authors also do not make any statements regarding the brain activity patterns they describe at different frequency bands. Are these the same or different between these tasks and earlier studies. What brain structures are they, what functions do they serve?

4) How do these results match/or not with the large literature on the topic using fMRI?

Overall, the paper is well executed, however, it would be stronger if its impact on the topic more generally is incorporated in the discussion.

Reviewer #3: The study by Mulders and colleagues investigates steady-state responses to heat and cold stimuli. The stimuli are applied in oscillations with 0.2 Hz. The authors conducted 2 experiments. In the first (behavioural) experiment, the relationship between stimuli and perception was explored; e.g. the authors found a delay between stimulation and perception.

In the second experiment EEG was recorded. The authors found various effects mainly at central and frontocentral electrodes.

The study is well-conducted in every aspect and I would very much suggest a publication in Plos One. However, I have quite a few points, which I would like to be considered. All of them are rather minor.

I would suggest rewriting the entire introduction. There is too much unrelated information on other techniques (fMRI), study designs (tonic, phasic) etc. that is not needed to understand the present study. Please, keep it short and your focus on EEG and SSRs. If required, mention these studies in the discussion. Based on the introduction, I would not know why the study needed to be done.

In my view, the main aspect of the study is the EEG time-frequency findings. I don’t understand why they are pushed to the margin of the manuscript. Perhaps, make a second publication on the results of the time-frequency analysis.

There are mostly within-subject comparisons. I think it would have been better to tailor the stimulus temperature to the individual.

The duration of the trials is very short (~75s). This has been compensated by a repeated stimulation (12 trials). The first part of the EEG responses might be attributed to saliency as the responses tend towards zero at the end. In my view, the end of the trials is the section where it starts to get interesting. I would suggest the authors to conduct a further experiment with one long trial, perhaps with adapted stimulus intensity.

For the EEG experiment, a contrast heat pain vs cold pain and heat vs cold would have been better.

I did not understand whether the authors always controlled for multiple comparisons. I would suggest considering Monte-Carlo simulations.

It would be interesting to see the individual time courses (habituation, delay) of the subjects. Rather than comparing the first vs second or last cycle I would suggest to fit the data of all 15 cycles to a function. The individual function parameters represent different aspects and can be tested against zero.

CWT does not to be explained.

The artefact correction is not sufficient. I would suggest considering ICA for eye movement and muscle artefacts.

The selection of only FCz electrode is not justified, particularly for alpha and beta. Include all electrodes in the main analysis part. There are tools available to compare topographies, such as TANOVA (included in RAGU software, http://dx.doi.org/10.1155/2011/938925)

Reviewer #4: Summary

In order to characterize the temporal dynamics human responses to a periodically modulated sustained thermal stimulus, the authors present experimental data on the use of a novel contact thermal stimulator with multiple thermal contact points to study cutaneous thermosensation and associated elicited brain activity.

The authors present two experiments on human subjects in which noxious heat or non-noxious cold stimuli to the skin were applied with a sinusoidal temperature profile at a frequency of 0.2 Hz, using several thermal stimulator modes of operation, including synchronous activation of all five thermal zones of the stimulation probe, alternating activation of two out of five thermal zones or activation of a fixed subset of two out of five thermal zones. In the first experiment, subjects were asked to continuously rate the perceived intensitiy of the stimulation and in the second experiment, the EEG of the subject is recorded and analyzed to estimate the amplitude at 0.2 Hz and higher harmonics.

From the results, the authors conclude that both sinusoidal heat and cold stimulation elicit sensations with periodically varying perceived intensities and also elicit EEG brain activity at the stimulus frequency of 0.2 Hz and higher harmonics, that intensities and brain responses habituate over time, but that these effects differ between heat and cold stimulation and between fixed and alternating stimulation modes.

Opinion

In my opinion, the authors present an original experimental study in which a welcome exploration is performed into the feasibility of using multi-zone thermal stimulation combined with a frequency tagging technique to study the processing of ‘tonic’ thermonociceptive stimuli. The manuscript is well written and accessible for the reader. Methods and Results are described and presented clearly. Although I recognize the merit of the methods and results as such, the authors may elaborate more on the rationale and relevance of this study and may take into account additional considerations as indicated below.

Major comments

1. In the introduction, the authors describe the peripheral structures and properties of thermonociception in detail and state that the link between peripheral activations and elicited brain activity and human perception are not fully understood yet (p3, line 18-21) and that (not explicitly formulated) questions are to be addressed (p3, line 23), some of which requiring sustained activation of thermonociceptor populations by tonic/periodic heat and cold stimuli. In my opinion, their rationale should also take the influence of central mechanisms like wind-up, sensitization or associated phenomena like temporal summation or offset-analgesia into consideration, as these are major factors in the transfer of peripheral to brain activity and also associated with tonic stimulation. This aspect should also be addressed when discussing the results.

2. The authors interpret the results on stimulus solely as temperature induced activation of thermonociceptors (e.g. cool-sensitive afferents) (p19, line 430), while a change in perceived intensity or elicited brain response can also arise from a temperature induced periodic reduced activation or modulation of already active afferents (e.g. warm sensitive) afferents. I am aware of the fact that this notion adds substantial complexity to the authors interpretations of the results in terms of thermonociceptor properties, but unless a rationale can be provided that justifies the exclusion of such modulation effects, the authors should be cautious and emphasize that their interpretations are preliminary.

3. Methods, p12, line 280: If I understand this correctly, you are every iteration replacing one cycle by frequency padding and computing the magnitude of noise-subtracted frequency spectrum. However, replacing a cycle of your signal by zero-padding would theoretically lower the magnitude at 0.2Hz, and by repeating this progress from 0 to all cycles you would expect a linear trend from the initial magnitude to zero respectively. Did you correct for this trend?

4. The results section is rather long in comparison to the discussion. You present a lot of results and perform a lot of statistical tests. However, not all results are used in the discussion section. Please leave out results and/or statistical tests that do not contribute to your message or are redundant. (Or extend the discussion if those results do contribute to your message).

Minor Comments

5. Methods, line 227: You subtract the average of eight neighboring frequencies. In the figure, the 0.2 Hz peak appears to be wider than just one frequency bin, in which case this approach would lower the SNR. Is 4-1-4 really the optimal width of you filter?

6. Results, line 320: I do not see this trend clearly in large-fixed and small variable, please refer to significance testing or refer from discussing a trend if it is not clearly visible and not significant.

6. PLOS authors have the option to publish the peer review history of their article (what does this mean?). If published, this will include your full peer review and any attached files.

Reviewer #1: No

Reviewer #2: Yes: A. Vania Apkarian

Reviewer #3: Yes: Enrico Schulz

Reviewer #4: No

---

## [Author Response · Author response to Decision Letter 0]

29 Jan 2020

The authors gratefully thank the Reviewers and Editors for their time as well as for their very interesting suggestions and relevant comments. The manuscript has been revised to address them as rigorously as possible. The document labeled ‘Response to Reviewers’ summarizes the changes and specifically explains how individual comments and suggestions of the Reviewers have been incorporated into the revised manuscript.

---

## [Decision Letter · Decision Letter 1]

31 Mar 2020

Dynamics of the perception and EEG signals triggered by tonic warm and cool stimulation

PONE-D-19-20397R1

Dear Dr. Mulders,

We are pleased to inform you that your manuscript has been judged scientifically suitable for publication and will be formally accepted for publication once it complies with all outstanding technical requirements.

With kind regards,

Jan R. Buitenweg

Guest Editor

PLOS ONE

Additional Editor Comments (optional):

Being invited as the guest editor, I participated as a reviewer for the initial evaluation of this manuscript.

Reviewers' comments:

Reviewer's Responses to Questions

**Comments to the Author**

1. If the authors have adequately addressed your comments raised in a previous round of review and you feel that this manuscript is now acceptable for publication, you may indicate that here to bypass the “Comments to the Author” section, enter your conflict of interest statement in the “Confidential to Editor” section, and submit your "Accept" recommendation.

Reviewer #3: All comments have been addressed

2. Is the manuscript technically sound, and do the data support the conclusions?

Reviewer #3: Yes

3. Has the statistical analysis been performed appropriately and rigorously? 

Reviewer #3: Yes

4. Have the authors made all data underlying the findings in their manuscript fully available?

Reviewer #3: Yes

5. Is the manuscript presented in an intelligible fashion and written in standard English?

Reviewer #3: Yes

6. Review Comments to the Author

Reviewer #3: All of my comments have been adressed. I fully recommend the publication of the manuscript at PLOS ONE.

7. PLOS authors have the option to publish the peer review history of their article (what does this mean?). If published, this will include your full peer review and any attached files.

Reviewer #3: Yes: Enrico Schulz

---

## [Editor Report · Acceptance letter]

3 Apr 2020

PONE-D-19-20397R1 

Dynamics of the perception and EEG signals triggered by tonic warm and cool stimulation 

Dear Dr. Mulders:

I am pleased to inform you that your manuscript has been deemed suitable for publication in PLOS ONE. Congratulations! Your manuscript is now with our production department. 

With kind regards,

on behalf of

Dr. Jan R. Buitenweg 

Guest Editor

PLOS ONE